# Applications of Solution NMR Spectroscopy in Quality Assessment and Authentication of Bovine Milk

**DOI:** 10.3390/foods12173240

**Published:** 2023-08-28

**Authors:** Dilek Eltemur, Peter Robatscher, Michael Oberhuber, Matteo Scampicchio, Alberto Ceccon

**Affiliations:** 1Laimburg Research Centre, Laimburg 6—Pfatten (Vadena), 39040 Auer, Italyalberto.ceccon@laimburg.it (A.C.); 2Faculty of Agricultural, Environmental and Food Sciences, Free University of Bozen-Bolzano, Piazza Unversità 5, 39100 Bolzano, Italy

**Keywords:** milk, nuclear magnetic resonance, metabolomics, chemometrics

## Abstract

Nuclear magnetic resonance (NMR) spectroscopy is emerging as a promising technique for the analysis of bovine milk, primarily due to its non-destructive nature, minimal sample preparation requirements, and comprehensive approach to untargeted milk analysis. These inherent strengths of NMR make it a formidable complementary tool to mass spectrometry-based techniques in milk metabolomic studies. This review aims to provide a comprehensive overview of the applications of NMR techniques in the quality assessment and authentication of bovine milk. It will focus on the experimental setup and data processing techniques that contribute to achieving accurate and highly reproducible results. The review will also highlight key studies that have utilized commonly used NMR methodologies in milk analysis, covering a wide range of application fields. These applications include determining milk animal species and feeding regimes, as well as assessing milk nutritional quality and authenticity. By providing an overview of the diverse applications of NMR in milk analysis, this review aims to demonstrate the versatility and significance of NMR spectroscopy as an invaluable tool for milk and dairy metabolomics research and hence, for assessing the quality and authenticity of bovine milk.

## 1. Introduction

Nuclear magnetic resonance (NMR) spectroscopy has proven to be a powerful tool for the analysis of milk and dairy products, given its unique ability to simultaneously detect and identify small molecules within complex matrices, such as milk [1]. NMR was initially employed by Odeblad and Westing in the 1950s to investigate milk properties and allowed the identification of distinct components of human milk: lactose from milk plasma and milk fat [2]. Since then, NMR spectroscopy has been extensively utilized in various studies, providing valuable insights into the composition, structure, and properties of milk, including the content of key components such as fat, protein, lactose, and minerals. One of the main advantages of NMR spectroscopy in milk analysis is its non-destructive nature, allowing for repeated measurements on the same sample over time and facilitating quality assessment throughout the production process [3]. Additionally, its inherent quantitative nature allows it to be used for quantitative metabolomic analysis of milk.

Metabolomics is an emerging tool in milk research; it gives valuable information about the health status and feeding system of the cow, the milk technological process and shelf-life, and eventually the nutritional quality of the milk. It offers reliable indicators and biomarker compounds that can be used for milk quality assessment and authentication. Over the past two decades, NMR has emerged as one of the most commonly used analytical techniques in milk metabolomic research, alongside gas chromatography coupled with mass spectrometry (GC-MS) and liquid chromatography coupled with single-stage mass spectrometry (LC-MS) [1]. Moreover, NMR-based milk metabolomics is continuously evolving as a promising approach offering high-throughput and comprehensive coverage of metabolites [4,5,6,7,8].

This review aims to highlight the various applications of NMR in the quality assessment and authentication of bovine milk, with particular consideration of metabolomics platforms. While previous literature reviews have covered the applications of NMR in milk and dairy products [4,9] as well as NMR-based metabolomics in their authenticity and quality assessment [5,6,7,10,11], this review places significant emphasis on the most recent advances and applications of NMR, particularly in bovine milk studies. The review will provide a comprehensive summary of commonly used NMR methodologies and NMR data analysis applied in the analysis of bovine milk, along with representative examples from related application fields. Although the primary focus of the review will be on high-resolution NMR, significant examples of low-resolution NMR will also be introduced as a promising tool in milk analysis at the industrial level. Figure 1 outlines the workflow of the review.

## 2. NMR Principle

Nuclear magnetic resonance (NMR) spectroscopy is considered among the most powerful analytical techniques used to determine the physical, chemical, and biological properties of matter. It allows for the provision of atomic-level information on molecules by measuring the interaction of external magnetic fields with the NMR active nuclei of molecules placed in a strong static magnetic field (B_0_). In fact, only certain naturally occurring nuclei have intrinsic properties that allow them to be used (and observed) in NMR analysis. Quantum mechanics tells us that those nuclei possess a nuclear spin angular momentum ($\vec{I}$) characterized by a nuclear spin quantum number (I). Only nuclei with nonzero nuclear spin quantum numbers, I ≠ 0 (meaning isotopes with an odd number of proton and/or neutrons) possess a nuclear magnetic moment, $\vec{\mu }$, that enables alignment in an external static magnetic field (similar to tiny bar magnets) and the ability to absorb an applied radiofrequency radiation (RF), thus allowing NMR measurements. The most commonly used NMR-active nuclei are those with a half-integer spin quantum number, especially I = ½. Fortunately, most common elements found in living organisms have at least one of the isotopes with spin number I = ½ (i.e., ^1^H, ^13^C, ^15^N, and ^31^P), thus allowing NMR applications in a wide range of biological samples. 

For spin I = ½ nuclei, the application of an external magnetic field B_0_ aligns the nuclear magnetic moment vector, $\vec{\mu }$, with or against the B_0_ field, corresponding to low energy or high energy spin states, respectively. The difference between the two energy levels is: ∆E = γhB_0_/2π,(1)

Linearly depends on the external magnetic field applied, B_0_ as well as on the gyromagnetic ratio, which is characteristic of each NMR active nucleus. The Boltzmann distribution defines the difference in population between the two energy levels as:N_high_/N_low_ = e^(^ −^∆E/kBT)^,(2)
where N_high_ and N_low_ are the populations of the upper and lower states, respectively, ∆E is the difference between the two energy levels, kB is the Boltzmann constant, and T is the temperature expressed in Kelvin. As for other spectroscopic methods (i.e., UV–vis), the amount of signal observed in NMR Spectroscopy is related to the difference in population between the two energy states. However, differently from UV–vis spectroscopy, at room temperature (T = 298 K), the upper and lower energy spin states are almost equally populated in NMR spectroscopy, with only a very small excess in the lower energy state, explaining NMR’s inherent low sensitivity compared to other techniques. 

For practical NMR applications, it is cumbersome to think in terms of individual spins and nuclear magnetic moment vectors ($\vec{\mu }$). Instead, it is more convenient to refer to the sum of all $\vec{\mu }$ in the sample, corresponding to the net magnetization (M). At equilibrium, M is aligned with the main magnetic field (B_0_). Irradiation of the sample with an RF pulse orthogonal to the main B_0_ tilts the magnetization M toward the xy plane. In quantum mechanics, irradiation of the sample with an RF pulse (generally applied for a few microseconds) of frequency (in Hz): v_0_ = γB_0_/2π,(3)
which determines the transition of spin between the energy levels (as indicated in Equation (1)). The energy absorbed by the nuclear spins induces a voltage that can be detected by a suitably tuned coil of wire, amplified, and the signal displayed as a free induction decay (FID). By converting the time-domain signal (FID) into a frequency-domain signal using the Fourier transform (FT), the FT provides a high-resolution NMR spectrum that contains valuable information about the molecular structure and dynamics of the sample. 

In the absence of any further perturbing RF pulses, two relaxation mechanisms, spin–lattice (T_1_) and spin–spin (T_2_) relaxations, eventually return the spin system to thermal equilibrium. The two mechanisms should not be confused and are profoundly different: If the former represents how quickly the net magnetization M returns aligned to B_0_ (i.e., the system recovers to its ground state), the latter refers to the progressively decaying magnetization M due to dephasing on the xy plane. Both types of relaxation can be measured on several NMR active nuclei by widely used and implemented relaxation-based NMR experiments and provide useful information for molecular structure elucidation and dynamic analyses.

In the following paragraph, the commonly utilized NMR observables in the field of food science, and more specifically in the characterization of milk by NMR spectroscopy, are described.

The chemical shift (δ) determines the position of a signal in the NMR spectrum and conveys information about the nuclei’s chemical environment. The chemical shift is measured in parts per million (ppm) relative to an internal standard, such as tetramethylsilane (TMS) or trimethylsilylpropanoic acid (TSP), which is typically set at 0 ppm. Using ppm instead of Hz makes the chemical shift independent from the spectrometer’s operational frequency, facilitating cross-comparison of data obtained from various spectrometers with different magnetic fields.

Spin–spin scalar J coupling is the result of neighboring spins interacting through bonding electrons, causing signal splitting in NMR spectra. Even though J coupling makes NMR spectra more complicated, it is crucial for structural characterization since its strength depends on the distance and dihedral angle between nuclei. However, splitting due to the J coupling can be removed using “broadband decoupling”, with the benefit of simplifying the spectrum and an increase in the signal-to-noise ratio (SNR).

The area of a given NMR peak is proportional to the number of NMR active nuclei that give rise to the peak, thus making modern NMR spectroscopy a robust method (qNMR) for the quantification of specific compounds. For example, a peak generated by three equivalent protons in a methyl group (CH_3_) has an area three times greater than the area of a peak caused by a single proton. Thus, one of the greatest advantages of the NMR method applied in food studies is its accuracy in quantitative measurements of the sample constituents because the intensity of the NMR signal is directly proportional to the molar concentration of the constituent [12,13]. Additionally, measurements of two mechanisms involved in the relaxation of magnetization (spin–lattice relaxation, T_1_, and spin–spin relaxation, T_2_) and measurements of translation diffusion are based on the determination of signal area. 

In the context of this review, we will be mostly focusing on the characterization of milk by 1D and 2D-NMR experiments (see relative section). In the former, the NMR spectrum is typically displayed as a plot of signal intensity (y-axis) versus frequency (x-axis) (expressed as chemical shift and referenced as described previously) measured on one nucleus (generally ^1^H or ^13^C), whereas in the latter, cross peaks are defined by two axes (x, y) indicating the chemical shift on two different nuclei. Experiments conducted in two dimensions can either involve the same type of nuclei (homonuclear, i.e., ^1^H-^1^H correlation spectroscopy, COSY) or different types of nuclei (heteronuclear, i.e., ^1^H-^13^C heteronuclear single quantum coherence spectroscopy, HSQC). Two-dimensional experiments offer improved resolution compared to 1D-NMR since spreading out the spectrum in two dimensions has the advantage for large molecules in that it removes much peak overlap. Additionally, combinations of 2D-NMR experiments (such as ^1^H-^13^C heteronuclear multiple bond correlation spectroscopy, HMBC, and ^1^H-^1^H correlation spectroscopy, COSY) allow assignment of chemical shifts (i.e., identification of NMR signals attributable to specific atoms).

## 3. NMR Experiments on Milk

### 3.1. 1D ^1^H-NMR Experiments

One-dimensional ^1^H-NMR spectroscopy is the most preferred NMR method in milk analysis mainly due to the high isotopic natural abundance (∼99%) and short relaxation time of ^1^H nuclei, leading to higher sensitivity and faster data acquisition [1,4] compared to other NMR experiments. Moreover, ^1^H-NMR spectra provide quantitative and very detailed structural information about milk constituents; therefore, it is widely performed in metabolomic studies. However, the presence of scalar J coupling between neighboring protons and a narrow chemical shift window (usually from 0 ppm up to ∼15 ppm) leads to an overcrowded spectrum with significantly overlapping peaks . This phenomenon results in low resolution of the ^1^H-NMR spectra and therefore makes interpretation of the peaks difficult. Resolution is a crucial factor in NMR spectroscopy, and it refers to the ability to distinguish between closely spaced peaks in the NMR spectrum. Furthermore, ^1^H-NMR experiments carried out on spectrometers with higher magnetic fields resolved this peak overlap by increasing the resolution of the acquired ^1^H-NMR spectra [1].

^1^H-NMR has been performed on a very wide range of applications in bovine milk studies. It has demonstrated excellent results on milk chemical composition characterization [14], nutritional quality determination of milk [15,16], discrimination of milk among different feeding regimes [17] and animal species [18], the effects of processing treatments on milk quality and shelf life [19,20,21], and adulteration of milk powder [22,23]. Figure 2 shows an example of the 1D ^1^H-NMR spectrum of milk that contains about 296 milk metabolites and metabolite species. More detailed information concerning the study is given in Section 7.

### 3.2. 1D ^13^C-NMR Experiments

One-dimensional ^13^C-NMR spectroscopy is commonly used in the analysis of milk lipid composition since it provides better resolution compared to ^1^H-NMR [9], and thus it is more informative in the identification and quantification of lipids [4]. In fact, ^13^C-NMR has a broader chemical shift window (∼200 ppm) than ^1^H-NMR (∼15 ppm), and therefore it has fewer overlapping peaks in the spectrum and, as described before, consequently higher resolution [24]. However, it is well known that quantitative analysis (qNMR) of ^13^C signals presents some complexities, in particular, related to (a) low sensitivity due to the low isotopic natural abundance of ^13^C (1%) (compared to other 100% abundance nuclei as ^1^H, ^19^F, or ^31^P) that require long experiment time (~hours), (b) long T_1_ relaxation time of ^13^C nuclei, and consequently long recycling delays (D1). Additionally, the degree of signal enhancement caused by the nuclear Overhauser enhancement (NOE) is proportional to the number of hydrogens attached to each ^13^C (except for quaternary carbons, which do not contribute to the enhancement) [1,25].

Therefore, the application of ^13^C-NMR, particularly in large-scale metabolomic studies of milk, has been limited. Nevertheless, it has been used as a very promising tool in milk lipid studies to discriminate milk among animal species [26], among different feeding regimes (organic and conventional production) [27,28,29], and among geographical origins [30].

### 3.3. 1D ^31^P-NMR Experiments

One-dimensional ^31^P-NMR experiments allow high resolution and sensitivity due to the 100% natural abundance of the ^31^P nucleus [25]. ^31^P-NMR spectroscopy has proved to be an effective technique in milk analysis to study phosphorus-containing compounds such as phospholipids, phosphorylated carbohydrates, inorganic phosphate (Pi), and modified amino acids such as phosphoserine (SerP) and the residue of casein [9]. ^31^P-NMR has also been applied to study the structure of caseins and casein micelles in milk [31]. It can be used to quantify and distinguish the different phosphorylated compounds present in milk as well as the diphosphates added to milk as stabilizers [32]. However, quantitative ^31^P-NMR analysis has a significant downside due to the lengthy T_1_ relaxation of the phosphorous nucleus, thus requiring time-consuming experiments to reach full relaxation of the magnetization, which is crucial for precise signal integration.

^31^P-NMR spectroscopy in milk analysis provides valuable information about the phospholipid composition of milk, and therefore it has been successfully employed for bovine milk authentication studies. Indeed, ^31^P-NMR phospholipid fingerprinting [33,34] and the metabolite profile of phosphorylated compounds [31] in bovine milk and milk from other animal species have been widely studied. ^31^P-NMR spectroscopy was also employed to investigate the effects of heat treatments on milk mineral equilibria and on the interaction between minerals (Pi and Ca) and micellar casein [35]. In general, the structure and structural changes of milk proteins, specifically caseins and casein micelles, can be investigated by ^31^P-NMR. In fact, this method allows one to observe signals (although broad) from casein micelles since the observable resonances belong to the phosphate of SerP present only in the micelles. Figure 3 represents the ^13^C and ^31^P-NMR spectra of bovine milk with the assignments of the most representative milk metabolites.

### 3.4. Two-Dimensional (2D) NMR Experiments

2D-NMR can potentially cover the main drawbacks of the most commonly used 1D-NMR experiments in milk studies, where the analysis often suffers from spectral congestion due to its low spectral resolution and peak overlapping or extensive signal splitting due to J-coupling. Additionally, 1D-NMR may not always be sufficient for the quantification of complex molecular structures, such as macromolecules in milk. On the other hand, 2D-NMR experiments are very efficient in the identification of both known and unknown metabolites, while 1D-NMR experiments are mostly limited to the identification of known metabolites [1]. Therefore, 2D-NMR seems more suitable for milk metabolomic studies. However, it is important to note that data acquisition and processing of 2D spectra can be more time-consuming and require skilled NMR spectroscopists with expertise in interpreting the obtained multidimensional data as well as in conducting metabolite identification and quantification.

Two-dimensional NMR has been used in milk analysis mainly for structural elucidation [9,38] and more recently also for the quantification of milk constituents [39]. Among 2D-NMR experiments, heteronuclear single quantum coherence spectroscopy (HSQC) and heteronuclear multiple bond correlation spectroscopy (HMBC) are the most frequently conducted in milk studies. In the former, the chemical shift of ^1^H is correlated to the directly bound ^13^C nucleus, whereas the latter experiment gives correlations between ^1^H and ^13^C nuclei that are separated by multiple bonds, usually up to three. Since HMBC spectra eliminate one-bond C–H correlations, it can be used for the assignments of quaternary and carbonyl carbons that are not detected in HSQC [1]. Indeed, these two 2D-NMR experiments are commonly applied as complementary experiments in complete assignments and chemical structure elucidation. ^1^H-^13^C HSQC experiments can adequately resolve signals from more analytes since extra dispersion is gained when the second dimension, ^13^C, is introduced [40]. Note that single-quantum HSQC experiments are generally preferred over multi-quantum HMQC experiments since they allow higher resolution on the second dimension despite being more susceptible to losses in SNR [41] An example of the ^1^H-^13^C HSQC spectrum of whole bovine milk with the most representative metabolite assignments is shown in Figure 4.

Two-dimensional heteronuclear NMR has been applied in milk studies for identification and quantification of milk constituents [40] and milk fatty acid content, aiming to overcome the sensitivity and resolution limitations of 1D-NMR in the detection of minor molecules [34]. Thus, it showed good results on bovine milk studies of authentication [42], geographical origin [43,44] and milk technological properties [45,46] . Although rare, the use of ^1^H-^31^P and ^1^H-^15^N HMBC was also reported in the literature for the assignment of NMR signals for lecithin [36].

## 4. Advantages and Disadvantages of NMR in Milk Studies

NMR spectroscopy represents a robust, cost-effective, and versatile technique for providing quantitative information on the metabolic profile at the molecular level. NMR analysis in milk studies does not require extensive sample preparation or chemical derivatization prior to analysis [47,48]. This unique advantage of NMR allows quick analysis, from a few minutes to a few hours for 1D ^1^H and 2D-NMR experiments, respectively. Additionally, the nondestructive nature of NMR allows repeated experiments with a high level of reproducibility as well as the possibility of performing further experiments on the same samples using additional analytical techniques [49]. 

Furthermore, NMR became an ideal technique in untargeted metabolomics over other techniques due to its unique strength to identify unknown compounds in milk that can be potentially identified as biomarkers in studies of milk authentication. Thus, it allows for identifying several compounds in milk, such as sugars, organic acids, vitamins, nucleotides, and aromatic compounds [6] that are less tractable in mass spectrometry-based methodologies. Moreover, the minimal sample preparation, without the need for separation steps such as liquid chromatography, allowed direct and reliable determination of analytes in milk samples by NMR and therefore facilitated the application of NMR, especially in large-scale metabolomic studies. Altogether, these unique strengths of NMR propose this technique as a complementary tool to mass spectrometry (MS), which has been widely performed in milk metabolomics coupled with liquid chromatography (LC-MS) or gas chromatography (GC-MS) [1,6].

However, there are several downsides to NMR spectroscopy. The most significant limitation of NMR is its inherently lower sensitivity compared to MS-based techniques. This is a consequence of the low spin polarization of the nuclei being studied (as described previously) as well as the low natural abundance of some nuclei of interest (for example, ~1.1% for ^13^C). The limit of detection of LC-MS and GC-MS methods is usually considered 10 to 100 times lower compared to NMR [1]. Moreover, even though NMR analysis is generally cheaper than MS analysis (mostly due to the minimal sample preparation steps required), high-field NMR spectrometers are expensive. It is mainly because of the associated running costs (i.e., periodic replenishment of the cryogenic liquids nitrogen and helium, necessary to keep the magnet at ~−271 °C) and the high investment costs of the instruments [6,11].

In addition to these instrumental challenges, the application of different extraction protocols and solvents or different data processing tools precludes any direct comparison of NMR spectra acquired on the same set of foodstuff samples, impeding any further statistical analysis of the NMR data [8,50]. The application of NMR in food analysis is relatively new; therefore, an international collaboration on standardization of data processing, software tools, and database maintenance is still needed. For instance, NMR-based metabolomics is far from having a comprehensive database; in fact, spectral libraries on food are limited with respect to MS-based metabolomics. Table 1 summarizes the advantages and disadvantages of widely used NMR methodologies for the analysis of bovine milk.

## 5. Experimental Conditions

The experimental workflow, from sample collection to data analysis, for the application of NMR spectroscopy in the study of milk content and composition is described here in detail. These are key steps to follow in the experimental setup to obtain high throughput and reproducible results. A summarized description of the experimental conditions is reported in Table 2.

### 5.1. Sample Collection and Pre-Treatment

Between collection and sample preparation, milk samples are generally stored at −80 °C. It is recommended to conduct an NMR analysis on the samples immediately after thawing. Even though storing samples at −80 °C after undergoing freeze–thaw cycles did not result in any noticeable impact on the NMR outcomes, it is not the ideal method for ensuring accurate analysis [1].

The quality of NMR experiments is directly affected by sample preparation, which makes it one of the crucial steps for a complete NMR analysis [60]. Several studies have demonstrated that sample preparation has a significant impact on the metabolite profile as well as on the spectral quality [13,29,61,62]. For instance, a study noted that methanol extraction protocols are more efficient concerning both spectral quality and metabolite extraction compared to ultrafiltration, skimming, and unprocessed milk protocols [62]. Another study showed the importance of optimizing sample preparation in milk lipid extraction (i.e., the extraction method, the ratio of the solvents used in the binary mixture, and the volume and quantity of the sample) to achieve an efficient extraction yield [29]. In fact, the application of accurate sample preparation protocols can ensure a more straightforward NMR output, especially for large-scale metabolomic studies where a large number of samples have to be analyzed. Overall, successful sample preparation should produce good-quality spectra with high SNR, low inter-sample chemical shift variability, and the highest number of metabolites detected in the sample [63].

However, the presence of sugars or other macromolecules such as lipids and proteins in milk is challenging due to different concentrations and polarities, which have an impact on the NMR analysis. These molecules give rise to broad signals in NMR spectra, which lead to overlapping or masking the signals of low-concentration metabolites, thus hampering accurate identification and quantification [6]. Therefore, in studies focusing on secondary milk metabolites, macromolecules are typically removed through sample preparation methods. The commonly used method for this purpose is precipitation. Proteins and lipoproteins are precipitated with methanol and ethanol, and lipids with chloroform [13]. Overall, these methods aim to obtain a milk extract with the highest recovery of metabolites and the highest purity from proteins and lipids [6].

In NMR-based milk metabolomic studies, several further sample preparation protocols have been described in the literature. These protocols describe the use of raw milk without any pre-treatment or additive [36,38], freeze-dried milk [29,43,54]or milk powder to be reconstituted prior to NMR analysis [17,18,64].

A quick approach for fatty acid profiling of milk samples requires the use of milk fat extract. Folch and Bligh–Dyer techniques are the most widely used milk lipid extraction methods, based on a chloroform:methanol:water mixture with a chloroform:methanol ratio of 2:1 (*v*/*v*) for extraction [54]. After the phase separation, the lower (organic) layer containing milk lipids is collected and evaporated under nitrogen [20].

The aqueous fraction of milk samples is widely studied in NMR analysis. Skimmed milk was obtained by centrifugation [17,21,25,64], by filtration (using a 10 kDa centrifugal filter unit) [14,22] or by ultra-filtration (using a 3 kDa centrifugal filter unit), which removes both lipid and protein compounds [23,24,26]. Alternatively, chemical precipitation of milk fat was achieved by the addition of chloroform [27,28], deuterated chloroform [32], and dichloromethane [29,30,31] followed by centrifugation. Aqueous fractions were also obtained with ultrafiltration [23,24,26], ultracentrifugation at 74,200× *g* for 60 min at 4 °C, followed by ultrafiltration [29,30], ethanol addition [25,33], or methanol addition [21] for protein denaturation and followed by a centrifugation step. The supernatant is generally dried under a stream of nitrogen and can be stored at −80 °C until further use. 

### 5.2. NMR Sample Preparation and Experimental Setup

NMR analysis typically involves a solution of the sample in an appropriate NMR solvent. In the case of ^1^H-NMR, a deuterated solvent is used to facilitate locking and shimming and suppress the large solvent signal, which otherwise could dominate the NMR spectrum [65]. The NMR solvent usually contains an internal reference mainly for (a) chemical shift referencing of ^1^H, ^13^C, and ^31^P nuclei and (b) allowing quantitative measurements. The choice of the internal reference strongly depends on the NMR solvent used. The most common internal references for polar solvents such as deuterium oxide (D_2_O) are 3-(trimethylsilyl) propionic-2,2,3,3-d4 acid, sodium salt (TSP), and 4,4-dimethyl-4-silapentane-1-sulfonic acid (DSS), whereas tetramethylsilane (TMS) is an ideal internal reference for non-polar solvents such as deuterated chloroform (CDCl_3_) [5].

Therefore, (1) milk fat extract obtained as described previously is dissolved in CDCl_3_ containing TMS as a chemical shift and intensity reference suitable for NMR measurements. Alternatively, (2) upon protein and/or fat precipitation, skimmed milk samples are prepared in a D_2_O phosphate buffer solution containing TSP as an internal reference. Finally, adequate volumes (at least 600 µL in a 5 mm tube to ensure good shimming and lineshape and consequently good spectral quality) of samples are transferred into the NMR tube for analysis. Furthermore, NMR samples can be repeatedly analyzed, stored under controlled conditions to avoid sample degradation, and reanalyzed to validate or confirm earlier findings [1].

An internal reference should have the following aspects: No volatility, complete solubility in the solvent, no chemical reaction with the solvent and/or sample, and no overlapping signal (at least one peak should be identified). For example, TSP and DSS have been shown to interact with proteins and fatty acids in the sample, and therefore they must be carefully evaluated, especially in quantitative NMR analysis [34]. If CDCl_3_ is used as the main solvent, the capillary of the NMR tube should be sealed prior to NMR analysis, e.g., with parafilm, since CDCl_3_ is volatile (boiling point at 60.9 °C) and evaporation can preclude quantitative measurements. In addition, it is essential to adjust the pH of the milk sample to prevent any possible chemical shift variations in the NMR spectra caused by slight pH differences between the samples. This issue can be solved by carrying out all NMR experiments in phosphate buffer (usually in a range of pH ~7.0 to 7.4) [35,36] and by further optimization during data analysis using an appropriate spectral alignment method [37].

#### NMR Experimental Setup for Quantitative NMR (qNMR)

Quantitative NMR (qNMR) spectroscopy is a straightforward analytical method with simultaneous identification and quantification of more than one analyte in a complex mixture in a relatively short experimental time (typically a few minutes for 1D ^1^H experiments) [66]. NMR-based milk metabolomics requires the acquisition of qNMR spectra. In milk analysis, ^1^H-qNMR is the most widely applied method since it provides high-throughput quantitative information on milk constituents with high accuracy. 

The absolute concentration of the component of interest can be directly calculated by the ratio of the peak area of the component of interest and the peak area of the reference material. Besides internal reference, which is most commonly used in NMR analysis, chemical or electronic external reference (acquired under identical conditions with the sample in different NMR tubes) or residual peaks from the solvent can be used as a reference for quantification [67]. Moreover, the reference material in qNMR can also have a different chemical structure from the analyte studied since there is no requirement for identical reference standards, such as in chromatography-based techniques [48,60].

For qNMR spectra, the following experimental conditions during spectra acquisition have been most widely used in the literature: 90° excitation pulse and repetition time (i.e., the sum of the FID acquisition time and any additional relaxation delay inserted prior to any RF pulse) at least five times the longest T_1_ to ensure sufficient relaxation of the magnetization [12,24]. This strategy produces the highest SNR in a single scan. Additionally, the length of the 90° pulse, the acquisition time (i.e., the length of the FID), the receiver gain, and the quality of shimming and tuning, all significantly influence the overall quality of NMR spectra [68].

### 5.3. 1H-NMR Spectra Acquisition: Water Signal Suppression

The quality of ^1^H-NMR spectra in NMR-based metabolomics significantly depends on the water suppression scheme, which is considered one of the critical steps of ^1^H-NMR signal acquisition. For instance, the presence of a residual water signal in the ^1^H-NMR spectra might overwhelm the detection of less concentrated milk metabolites that resonate close to H_2_O as well as determine spectral baseline distortions [69]. Although the water content can be mostly eliminated by lyophilization, this process elongates the sample preparation time [25], and the remaining water in the milk sample is in most cases not negligible for the quality of the NMR spectra [12].

Modern NMR instruments allow water signal suppression by implementing several pulse schemes; the most common is presaturation [69]. WATERGATE and Excitation Sculpting [70,71], and one-dimensional nuclear Overhauser effect spectroscopy (1D-NOESY) [72]. Even though there are several options available, the 1D-NOESY sequence has emerged as the most commonly used in NMR-based metabolomic studies [61]. Using a combination of presaturation and excitation pulses, 1D-NOESY allows minimal baseline distortion in the NMR spectra compared to other methods [25,73]. Various NMR data processing software such as TopSpin (Bruker, San Jose, CA, USA), Delta (JEOL, Tokyo, Japan), Mnova (Mestrelab Research, Santiago, Spain), etc. [74] also allow removing the water signal post-acquisition. Each tool has a special algorithm with different efficiencies for water signal removal. Therefore, the most appropriate technique must be chosen depending on the focus of the research. Spectral processing steps such as the choice of zero filling, apodization, chemical shift referencing protocol, phase, and baseline corrections are of particular importance in NMR spectra treatments [68]. To ensure consistency and standardization of the process, it is essential that all spectra are treated in an identical manner and, in particular, if some steps are carried out manually, by the same operator. Nonetheless, processed NMR spectra may still suffer from chemical shift drift due to instrumental variations such as temperature and magnetic field oscillation, baseline distortion [63], or variations among samples such as differences in pH or ionic strength. While instrumental variations that result in an overall spectrum-to-spectrum chemical shift drift can be solved by using an internal reference peak, it is more difficult to handle sample-borne variations that result in local peak-to-peak chemical shift drifts [75].

## 6. Spectral Processing and Data Analysis

Following NMR spectra acquisition, proper spectral processing is applied (i.e., phase correction, zero-filling, baseline correction, and chemical shift referencing) to the NMR data. Subsequently, NMR data processing is required prior to multivariate data analysis to correct possible variables in the dataset and extract the most useful information for the study. Several studies have provided detailed guidelines and recommendations on NMR data analysis, covering spectral processing, data processing, and multivariate data analysis [68,75,76,77,78,79]. The present review will briefly highlight the essential steps of NMR data analysis to provide a comprehensive overview of NMR applications.

### 6.1. Spectral Processing

Phase and baseline corrections are the initial and most important steps in NMR spectral processing. Baseline correction is essential for removing baseline offsets and distortions caused by the imperfect magnetic field homogeneity of the NMR instrument. It is usually performed via semiautomatic tools with manual identification of relevant baseline regions followed by automatic completion of the remaining baseline correction [68]. Frequency domain correction methods have been more widely used compared to time domain methods because they provide more accurate correction by constructing the baseline curves in the spectra directly [80]. The quality of phase and baseline corrections can affect the accuracy of peak integration and quantification. These processing steps are critical, especially in metabolomic studies where low-concentration metabolites significantly contribute to the results. For instance, a proper baseline correction allows accurate assignments of low-intensity peaks that are sensitive to baseline distortions [80]. Moreover, manual phasing is often needed in metabolomics for a more accurate identification of low-intensity peaks [68]. Consequently, it is important to choose appropriate spectral processing parameters to obtain reliable and accurate results. 

Nowadays, spectral processing methods are applied automatically to minimize the influence of the operator on the results and simplify the processing of large sets of NMR spectra [81]. A plethora of software packages are available for spectral processing and further data analysis. Previously mentioned TopSpin (Bruker, San Jose, CA, USA) and Delta (JEOL, Tokyo, Japan) software, also used for NMR spectral acquisition, allow optimal spectral processing. Other common NMR processing software packages include NMR Suite (CHENOMX, Edmonton, AB, Canada), Mnova (Mestrelab Research, Santiago, Spain), and Amix toolkits (Bruker, San Jose, CA, USA) [76]. However, milk studies often require more sophisticated multivariate data analysis due to the complexity and variability of milk chemical composition. 

The NMR spectra of milk often have overlapping signals that will affect the accuracy of identification and quantification of studied compounds; therefore, conventional peak integration is mostly inadequate [82]. A more advanced spectral deconvolution approach has been alternatively applied for the integration of these peaks, disentangling overlapping signals into their individual spectral patterns [83]. Spectral deconvolution can be performed manually or automatically. Thus, there are various deconvolution tools using different quantification algorithms available for both 1D and 2D-NMR spectra [1].

### 6.2. Data Processing

NMR spectra are initially aligned to the internal reference in each spectrum to optimize the possible chemical shift drifts, as described previously. Further, a spectral alignment method is needed for an advanced correction of the peak shifting. Binning (also known as bucketing) is the most commonly used spectral alignment method. The binning process involves dividing the NMR spectrum into a set of equally sized intervals (between 0.01 and 0.05 ppm range) or bins, along the chemical shift axis [63,84]. The spectral intensity in each bin is then integrated to generate a set of bin intensities, which are adjusted (i.e., normalized) to achieve a common scale or baseline across the spectra and then used as input for further analysis such as multivariate statistical analysis, machine learning, or database searching. To optimize the binning process, it is important to choose an appropriate bin size that balances spectral resolution and information content and to carefully evaluate the effect of binning on the downstream analysis. In fact, the main drawback of this method is the loss of both spectral resolution and data that may be relevant to the study, due mainly to the uncontrolled data reduction [85]. Therefore, more sophisticated alignment methods such as dynamic time warping (DTW) [80], correlation optimized warping (COW) [86], and interval correlation optimized shifting (icoshift) [84] have emerged to obtain better results from NMR spectra. After NMR spectra are aligned, normalization is applied to correct the variations in signal intensities among the samples. These variations occur due to the possible differences in sample concentrations, dilutions, and instrumental signal instability, which are very common in the analysis of milk samples [29]. Normalization of NMR spectra refers to the process of scaling the spectral intensities to a common reference point to correct differences in sample concentration and instrument settings [12]. The most common method of normalization is based on total spectral area normalization, where the total area under the NMR spectrum is used as the reference point. This method assumes that the total number of protons is constant and that there are no major changes in the sample matrix or experimental conditions [33].

However, in the case of milk, normalization to total intensity would indeed normalize to lactose concentrations, which is the most abundant metabolite in milk [6,62]. This could lead to misleading results if the goal is to compare the relative concentrations of other metabolites in different milk samples. In such cases, alternative normalization strategies can be used to adjust for the differences in the metabolite concentrations. The standard approach is to normalize the concentration of an internal reference that is added to the sample before NMR analysis [87]. The internal standard should have a known concentration and ideally have minimal overlap with the sample metabolites in the NMR spectrum. Normalization to the internal reference can effectively adjust for variations in sample preparation, NMR measurement, and other experimental factors that could affect the spectral intensity.

Prior to data analysis, data scaling is essential to reduce the spread in the variables where the higher intensity peaks have a higher influence in the multivariate data models [88]. The data scaling allows for an increased contribution from low-concentration peaks in the model. This step is crucial in metabolomic studies to ensure accurate results. Among various scaling methods [6,88,89], Pareto scaling is commonly used in NMR-based milk metabolomics due to its ability to maintain the data close to their original values while reducing the misleading effect of large variances [29]. Thus, to achieve optimal results in NMR data processing, a combination of autoscaling, Pareto, and centering scaling methods is recommended for identifying the discriminative metabolites between two groups [90].

### 6.3. Metabolite Identification

Solution NMR spectroscopy allows for the identification, quantification, and detailed structural information of metabolites present in bovine milk. One-dimensional and 2D-NMR experiments are commonly employed to provide detailed information about the chemical shifts and coupling patterns of the metabolites. The metabolite assignments are confirmed through multiple experiments, including 2D-NMR experiments such as COSY, TOCSY, and HSQC previously described in Section 3.4. The chemical shift values obtained from the NMR spectrum are compared with databases and reference spectra containing known metabolite chemical shifts. The most common databases that provide valuable reference data for metabolite assignments are the Biological Magnetic Resonance Data Bank (BMRB) [91], Human Metabolome Database (HMDB) [92], the Madison-Qingdao Metabolomics Consortium Database (MMCD) [93], and NMRShiftDB2 [94]. Commercial software packages, such as Chenomx NMR Suite (Chenomx, Edmonton, AB, Canada) and KnowItAll Metabolomics (BioRad Corporation, Hercules, CA, USA), rely on standard spectral libraries and proprietary databases that are created and maintained by the respective companies.

### 6.4. Multivariate Data Analysis (MVDA): Unsupervised and Supervised Methods

NMR-based metabolomic datasets obtained from milk may contain a large number of samples and variables, making it challenging to extract meaningful information without the application of advanced multivariate data analysis (MVDA) tools. Unsupervised MVDA methods, such as principal component analysis (PCA) and hierarchical cluster analysis (HCA), are commonly employed in NMR-based milk metabolomics. These methods are used to identify the inherent patterns and groupings within the samples without prior knowledge of the sample class or treatment. Briefly, PCA can be used to reduce the complexity of large NMR datasets by identifying the most significant variations in the sample set and projecting the data onto a lower-dimensional space. HCA, on the other hand, groups samples into clusters based on similarities in their spectral profiles and can be used to identify outliers or subgroups within the data. Both methods not only reveal patterns and groupings within NMR-based milk metabolomic data, but they also provide valuable insights to predict the values of target variables of interest, such as milk composition, metabolic profiling, and processing quality [7,63]. Contrarily, supervised techniques rely on prior knowledge of class membership to construct classification models that can predict the class membership of unknown samples [88]. Supervised MVDA methods, such as partial least square discriminant analysis (PLS-DA) and orthogonal variants of PLS-DA (O-PLS-DA), are the most commonly applied data analysis methods in milk studies. 

In NMR-based studies on milk, PCA is usually performed prior to any other supervised MVDA methods as an initial exploratory tool to gain insights about the dataset [12]. PCA is often combined with PLS-DA, particularly in milk quality assessment and adulteration studies. A typical multivariate data analysis process in milk analysis follows a metabolic approach, where the metabolite content significantly differs between the sample groups according to the NMR spectra and is used as input variables into subsequent multivariate analysis [25,30,95]. Followingly, PCA loading plots are used to determine whether those characteristic variables identified in the NMR spectra can distinguish the sample groups. Variable selection is then performed to identify which variables contribute most to group discrimination. Supervised PLS-DA is subsequently conducted, and variable importance in projection (VIP) scores are calculated to determine the significance of the studied variables. Finally, milk metabolites with a VIP value higher than 1 are usually selected as potential biomarkers for distinguishing the sample groups [54]. Figure 5 shows the PCA score plot, PLS-DA score plot, and OPLS-DA corresponding color-coded plot from a study that investigated bovine milk adulteration in caprine milk [96]. In the study, four NMR signals originating from orotate, citrate, N-acetyl carbohydrates, and one unknown compound were detected as more abundant in bovine milk, and they were identified as potential biomarkers for differentiating the two groups and estimating the adulteration percentage in the bovine–caprine milk mixture. Overall, the multivariate model demonstrated clear clustering between the two groups.

## 7. NMR Applications on Bovine Milk: Quality Assessment

NMR spectroscopy has been widely applied in studies of milk quality assessment. Various studies have shown the capability of NMR in determining the composition of bovine milk [14,36], distinguishing milk from different animal species [26], different farms [44], and discerning milk from different feeding systems [17,95] Moreover, NMR analysis enables the evaluation of the nutritional value of milk [25,59], monitoring the milk quality during processing and storage [14,25,48,63], and assessing the technological properties of milk [45,46]. Table 2 provides a summarized description of the application fields of NMR studies on bovine milk, including the NMR experiments and types of samples used for the corresponding investigation.

### 7.1. Milk Composition

The pioneering study by Hu et al. [36] in 2004 holds a significant place in the literature as the first investigation to characterize the chemical composition of whole bovine milk using NMR spectroscopy without any additive or sample pretreatment. In the study, several milk metabolites were assigned to perform both one-dimensional (^1^H, ^13^C) and two-dimensional (^1^H-^13^C, ^1^H-^15^N, ^1^H-^31^P) NMR experiments. Notably, they identify two compounds, creatine and N-acetyl carbohydrates, for the first time in milk through NMR analysis. It was observed that the addition of 10% D_2_O assisted in adjusting the lock and shims, thereby improving the resolution and sensitivity of the NMR spectra. However, the study demonstrated that even without pretreatment, milk can be analyzed for the assignment and quantification of a large number of metabolites [36]. Additionally, the study revealed that non-homogenized milk showed broader signals in the ^1^H-NMR spectra compared to homogenized milk. This broadening effect was attributed to the presence of larger fat globules in non-homogenized milk, which results in signal broadening and reduced NMR signal sensitivity. This finding later influenced NMR-based milk analysis, allowing for the monitoring of milk fat globule size and stability, as well as the assessment of milk homogenization quality. In their following study published in 2007 [38], fatty acids and various milk compounds were successfully quantified by 2D-NMR analysis of whole milk with the addition of 10% D_2_O. The authors stated that the study has been reported as the first study using 2D-NMR for the quantification of food components [38]. Overall, these studies have contributed to the development of the application of NMR in milk studies.

Foroutan et al. [16] conducted a comprehensive study on the chemical composition of bovine milk, employing a multi-platform approach that encompassed targeted and quantitative metabolomics through NMR, LC-HRMS, LC-MS/MS, and inductively coupled plasma-mass spectrometry (ICP/MS). This study, conducted in 2019, reported the largest number of metabolites quantified in bovine milk at that time, including the identification of some new compounds in milk such as lysophosphatidylcholine (18:2), phosphatidylcholine (28:1), and triglyceride (48:3) [14]. In total, 296 milk metabolites and metabolite species were quantified and validated using both quantitative metabolomics and literature mining approaches. The authors successfully identified 2355 unique metabolite structures in milk, and the data are readily accessible through a web-accessible database called the Milk Composition Database (MCDB).

### 7.2. Milk Origin

The pioneering NMR technique has been successfully applied to distinguish milk types according to their animal species origin. Andreotti et al. [26] managed to differentiate bovine milk from sheep and goat milk through fatty acid composition using ^13^C-NMR spectroscopy combined with fuzzy logic analysis. Moreover, Garcia et al. [33] optimized a ^31^P-NMR fingerprinting approach (with validation on GC) to differentiate milk samples from different animal species (bovine, human, camel, and mare) through their phospholipid (PL) profiles. This method enabled the identification and quantification of multiple PLs in milk samples, facilitating the discrimination of milk according to their animal origin [33]. Bruschetta et al. [31] also highlighted the effectiveness of ^31^P-NMR fingerprinting in differentiating milk samples from various animal species (e.g., cow, sheep, goat, mare, and donkey). They observed similar ^31^P-NMR profiles among milk from the same animal species, irrespective of the breed, age, or lactation of the animal [32]. Moreover, Wei et al. [34] conducted a comparative study on the chemical composition of milk from five different mammals, including bovine, human, goat, yak, and donkey. They investigated the PL composition of milk samples using ^31^P-NMR and LC-MS, as well as the fatty acid composition using GC. The study revealed variations in the concentrations and composition of PLs and the size of fat globules among milk samples from different animal groups. Interestingly, human milk exhibited a fat globule size similar to that of bovine milk. These findings suggest that NMR analysis could be utilized to optimize infant formulas to more closely resemble human milk in terms of PLs and milk fat globule structure, thereby ensuring the nutritional quality of such formulas [34].

### 7.3. Milk from Different Feeding Systems

Recent studies have utilized NMR spectroscopy to investigate the relationship between milk cow feeding patterns and milk composition and quality. O’Callaghan et al. [51] conducted a study comparing pasture feeding systems to indoor feeding systems and their effects on the milk metabolome using ^1^H-NMR metabolomics combined with multivariate data analysis. In the study, milk samples were collected from three commonly practiced feeding systems in Ireland: Pasture perennial ryegrass, perennial ryegrass with white clover, and an indoor total mixed ration consisting of maize and grass silage. The milk samples from cows on the two different pasture feeding systems exhibited similar metabolomes throughout the entire lactation period. In contrast, a significant difference was observed between the milk samples from cows in the indoor silage feeding system and the pasture feeding system. For instance, milk samples from pasture feeding had a higher hippuric acid content, while milk from silage feeding had higher urea content [51]. The increased concentration of urea in milk can have a negative impact on milk protein quality, as it is the main source of non-protein nitrogen content in milk [51]. Thus, the authors stated that hippuric acid could serve as a biomarker for milk from pasture-based feeding regimes since its presence in milk is correlated with forage feeding. Conversely, Lanza et al. [17] reported a higher concentration of hippuric acid in milk from maize silage feeding compared to hay feeding. In the study, both the milk polar fraction and fatty acids of milk samples from three different feeding regimes were analyzed using ^1^H-NMR and GC-MS, respectively. Milk samples from lucerne hay feeding were found to have considerably higher levels of polyunsaturated fatty acids (PUFA) compared to those from maize silage or maize silage partially replaced with grass–legume silage feeding systems. Furthermore, the NMR milk polar metabolomic profile was found to be less sensitive to alterations in the cow’s diet compared to the milk FA profile. Indeed, the authors noted that the overall metabolomic profile of milk remained largely unaltered when maize silage was partially substituted with a grass–legume silage mixture. However, a significant change was observed only when maize silage was completely replaced with hay [17]. These studies highlight the reliability and precision of NMR in assessing the impact of feeding practices on milk quality.

### 7.4. Milk Technological Process

Recent NMR techniques have shown significant potential for monitoring milk quality and shelf life during various processing applications and subsequent storage. D. Zhu et al. conducted studies addressing two relatively unexplored aspects in the field: The impact of vat pasteurization [54] and the effect of freeze-drying [19] processes on milk quality under different storage conditions. Both studies employed untargeted metabolomics using ^1^H-NMR and MS-based techniques, followed by multivariate analysis. In the study on vat pasteurization, the authors observed no significant changes in the milk metabolite profile following the pasteurization process. However, during subsequent refrigerated storage (4 °C), the concentration of free fatty acids and some organic acids (such as succinic acid, ribonic acid, and galactonic/gluconic acid) increased, while the concentration of pantothenic acid (vitamin B5) decreased [54]. In the study on freeze-drying, by using ^1^H-NMR and ^13^C-NMR combined with MS, the authors monitored the effect of the freeze-drying process and different storage conditions (varying temperatures and durations) on the milk metabolome over a period of 224 days. Multivariate analysis of the combined data showed that the freeze-drying process had a slight effect on the content of milk metabolites. Additionally, stable metabolome profiles were observed in freeze-dried milk samples stored in the fridge (4 °C) or in the freezer (−20 °C). In contrast, significant changes in the content of some milk metabolites were detected in freeze-dried milk samples stored at room temperature (20 °C), including decreased concentrations of riboflavin, orotic acid, and acetyl carbohydrates, increased concentrations of fatty acids, threonic acid (an oxidized product of ascorbic acid), and uridine [19]. Indeed, the authors highlighted that both vat pasteurization and freeze-drying are effective and mild preservation methods that cause only minor changes in the milk metabolome [19,54].

Innovative processing applications and their effects on milk quality have also been studied using NMR. Lemos et al. [20] focused on the application of NMR-based metabolomics to investigate the effects of hyperbaric preservation on the quality of high-pressure pasteurized (HPP) milk. The study monitored the impact of storage on the metabolic profile of HPP milk stored at an uncontrolled room temperature (~20 °C) with three different pressure levels (50, 75, and 100 MPa) and a storage period of 40 days. For comparison, two control groups were stored at either room temperature or refrigeration (4 °C) under atmospheric pressure. Visual NMR spectra evaluation revealed significant differences between milk samples subjected to hyperbaric storage and the control groups, even after just three days of storage. From the analysis of the NMR spectra, milk samples stored at room temperature showed indications of spoilage, including increased levels of lactate and butyrate; therefore, they were not included in the data analysis. Moreover, milk samples stored under hyperbaric conditions showed broad peaks in both aliphatic and aromatic regions due to an increase in soluble protein levels over time, which was mainly attributed to HPP treatment. This increase in protein content was associated with higher protein digestibility and higher nutritional value. The authors emphasized the potential of NMR-based metabolomics as a powerful tool for monitoring the effects of hyperbaric preservation of HPP milk quality and shelf life [20]. In another study, Yang et al. [21] investigated the effects of single- and double-cycled hydrostatic pressure treatment on whole and skimmed milk samples with high bacterial loads. They utilized ^1^H-NMR for metabolic profiling, GC-MS for the analysis of fatty acids and volatile compounds, and sodium dodecyl sulfate-polyacrylamide gel electrophoresis (SDS-PAGE) for protein profiling. The study revealed that skimmed milk is more sensitive to high-pressure treatment compared to whole milk, indicating the possible baroprotective effect of milk fat on milk micelles. Additionally, multi-cycled hydrostatic pressure treatment was more effective in microbial inactivation compared to single-cycle treatment, resulting in better preservation of milk quality [21]. Overall, these studies conclude that NMR-based metabolomics can be used to monitor milk quality during processing and storage, making it a valuable tool for shelf-life studies of milk.

### 7.5. Milk Nutritional and Technological Quality

The NMR method has been successfully used for the evaluation of the nutritional and technological quality of bovine milk. Sundekilde et al. [46] investigated the relationship between the metabolic profile and technological properties of bovine milk by using NMR-based metabolomics. In the study, skimmed milk samples were classified based on their coagulation properties, which had previously been characterized as either good or poor-coagulating milk. The samples were analyzed using ^1^H-NMR and ^13^C-NMR spectroscopy combined with multivariate analysis. The authors found a positive correlation between the coagulation properties of milk and the concentrations of lactose and choline. On the other hand, they observed a negative correlation between carnitine and citrate and the coagulation properties of milk [46]. Therefore, the authors suggested that NMR-based metabolomics could be a rapid classification method to differentiate milk according to its coagulation properties. In a subsequent study, Sundekilde et al. [45] examined the correlation between milk metabolites and their protein content and how they affect the coagulation properties of milk. The study revealed that specific metabolites related to milk protein content, such as choline, creatinine, and carnitine, showed notable differences between non-coagulating and well-coagulating milk samples, indicating their potential use as indicators of milk quality in terms of coagulation properties [45].

The regulation of infant formula requires a detailed declaration of ingredients on the label, as the formulation plays a crucial role in the healthy growth of infants. An untargeted ^1^H-NMR approach was used by Zhao et al. [23] to determine the nutritional value and compliance of the infant formulas with regulatory requirements. The low-molecular-weight organic compounds in whole bovine milk and commercial infant formulas from different brands were investigated. The study revealed significant differences between milk and infant formulas, as well as the presence of some metabolites that were not declared on the label [23]. Moreover, the authors highlighted the importance of determining choline, creatine, and certain nucleotides/nucleosides by NMR, as these milk compounds have beneficial bioactivities for infants and hence can be used as marker compounds to validate the nutritional value of infant formula [23].

### 7.6. Milk Cow Health Status

The health and metabolic status of dairy cows strongly influence milk composition and, consequently, the quality of milk. Luangwilai et al. [15] conducted a study in which they characterized and differentiated 46 milk metabolites in raw milk obtained from healthy cows, as well as cows with subclinical and clinical mastitis, using untargeted ^1^H-NMR. Mastitis inflammation in cows, which leads to an extremely high somatic cell count, was found to be correlated with variations in the milk metabolome. This resulted in significantly increased concentrations of various free amino acids and organic acids in milk. Moreover, the authors reported that increased levels of alanine, valerate, and N-acetylglucosamine in milk can be used as new potential biomarkers for diagnosing mastitis in cows [15]. In another study, Sunds et al. [52] investigated glycosidase enzymatic activities in mastitis milk and their effects on milk composition using untargeted ^1^H-NMR metabolomics. The presence of certain glycosidases was associated with mastitis inflammation, which had a substantial impact on the milk metabolite profile. The enzymatic activities led to increased levels of free sugars and decreased levels of lactose. Additionally, higher concentrations of fat, protein, and free amino acids were observed, with the latter being attributed to the proteolytic activities of microorganisms due to mastitis [52]. Followingly, the mechanisms of mastitis inflammation were explained by C. Zhu et al. [16] through milk metabolomics using untargeted ^1^H-NMR. The study provided quantitative information on milk metabolites by comprehensively analyzing both milk and serum samples. They characterized and quantified 54 milk metabolites; a greater number compared to previous reports in the literature. The variations in metabolites between serum and milk samples were analyzed to understand the metabolic pathway of mastitis in milk [16]. In a different study, Xu et al. [53] investigated the biological pathways of negative energy balance in dairy cows by using ^1^H-NMR combined with LC-MS through integrated analysis. They focused on the metabolite profile of milk serum obtained from cows in the second week of lactation. Milk samples from cows experiencing negative energy balance showed various alterations in the milk metabolic profile. The authors suggested that the combination of NMR with LC-MS as a complementary technique can provide integrated and reliable information about the energy status of dairy cows and, consequently, the nutritional value of milk [53].

## 8. NMR Applications on Bovine Milk: Authenticity

NMR has emerged as a reliable method for verifying the authenticity of bovine milk. It has been employed as an analytical tool for assessing the authenticity of organic milk [28,29], determining the composition of reconstituted milk [18,55], detecting adulteration in milk powder [22,39], and identifying milk adulteration with melamine [56]. In addition, time-domain NMR has been applied in the assessment of milk quality and authenticity [58,59]. Table 2 offers a concise overview of the application of NMR studies in this field.

### 8.1. Bovine Milk Authenticity

Erich et al. studied the fat composition of milk to assess the authenticity of organic milk. In the study, ^1^H-NMR and ^13^C-NMR methods were used in combination with stable isotope-ratio mass spectrometry (IRMS) and gas chromatography (GC). Visual NMR spectra comparison combined with chemometric models revealed that organic milk exhibited stronger signals on the bis-allyl methylene groups of α-linolenic acid, the carboxyl group of butyric acid, and the long-chain fatty acids than conventional milk samples [28]. Moreover, chemometric analysis (PCA and various classification methods) applied to NMR data combined with stable-isotope data of milk protein and fat (obtained from IRMS) and α-linolenic acid content (quantified using GC) allowed to differentiate organic milk from conventional milk samples [28]. Tsiafoulis et al. [29] analyzed the lipid fraction of lyophilized organic and conventional milk samples using both targeted and untargeted NMR approaches, with a particular interest in the potential health benefits of organic milk for human consumption. From 1D and 2D-NMR spectra, significantly higher concentrations of conjugated linoleic acid (CLA), linoleic and α-linolenic acids, and overall unsaturated fatty acid content were measured in organic milk. Those molecules are associated with the better nutritional value of organic milk, providing human health benefits upon consumption [29].

Renou et al. [30] conducted a study to determine the source of milk from two distinct regions—mountainous and plain—where cows were fed with a combination of pasture and silage. Milk fatty acid composition was studied using ^13^C-NMR spectroscopy, whereas the milk aqueous fraction was studied by IRMS as a complementary method. Discriminant analyses considering two factors—geographic origin and feeding—were performed based on five variables: Relative proportions of monounsaturated fatty acids (MUFA), polyunsaturated fatty acids (PUFA), saturated fatty acids (SFA), ^18^O/^16^O, and deuterium/hydrogen ratios. The study revealed that the level of PUFA in mountain milk was considerably higher than in plain milk, while MUFA and SFA levels did not differ significantly between the two regions. As a result, milk from cows that grazed on pasture had a higher concentration of MUFA compared to milk from cows fed silage. In addition to the impact of feeding practices on milk fatty acid composition, the study conducted by Renou et al. [30] found that oxygen isotope ratios were more influenced by geographical location. 

Furthermore, Sacco et al. [43] employed NMR in combination with IRMS to assess the authenticity of milk from Southern Italy. This innovative approach was compared to traditional spectroscopic and chromatographic techniques. The ^1^H-NMR spectra of milk samples from Southern Italy showed some differences when compared to milk from foreign countries, which were also confirmed from 2D-NMR molecular assignments. The ^1^H-NMR spectra of southern Italian milk samples displayed two peaks at 1.32 and 4.11 ppm, which represent CH- and CH_3_-signals of lactic acid, respectively. In contrast, these peaks were absent in foreign milk samples. Thus, foreign milk showed a higher glycerol and sugar content, as well as a different amino acid composition, compared to Southern Italian milk. These differences in milk metabolites were attributed to distinct feeding systems. Additionally, a multivariant statistical analysis using the variables from the most discriminating peaks and isotope ratios confirmed clear differentiation between milk samples from Southern Italy and those from foreign countries [43]. These studies revealed that NMR combined with IRMS provides a more precise approach for geographical origin determination and authenticity assessment of milk compared with traditional techniques [30,43]. Furthermore, Tenori et al. [44] made significant advancements in determining the geographical origin of milk using NMR-based metabolomics. Their approach successfully distinguished milk samples from different farms and brands within the same area and region. However, they emphasized the need to consider seasonal variations that can impact milk composition [44]. Overall, NMR can provide a fast and dependable way to detect the geographical origin of milk, which can be particularly relevant for EU-regulated geographically indicated milk products.

### 8.2. Bovine Milk Adulteration

NMR spectroscopy has been effectively used in adulteration studies to identify the presence of milk from various other sources in bovine milk. Lamanna et al. [18] conducted a study using ^1^H-NMR metabolomics to detect the amount of sheep milk in bovine milk as a means to detect adulteration. The study involved preparing a milk mixture of sheep milk and bovine milk at different concentrations. By analyzing the aqueous fraction of milk samples, the study identified spectral differences in metabolites such as citrate, lactate, and protein content, enabling the differentiation of pure bovine milk from pure sheep milk. Moreover, a combination of ^1^H-NMR metabolic profiles and multilinear regression statistical analysis allowed the determination of the relative concentration of each milk type in the milk mixture [18].

Furthermore, Li et al. [42] applied 1D and 2D-NMR experiments combined with chemometrics to distinguish between bovine, goat, and soy milk. The study identified specific metabolites that differentiate these milk types, such as higher levels of N-acetyl carbohydrates, ethanolamine, citrate, and lecithin in bovine milk compared to goat milk. On the other hand, lower levels of acetate, carnitine, and creatine were found in bovine milk. Additionally, the presence of D-sucrose was identified as a biomarker for the adulteration of bovine milk with soymilk [42]. This method proved effective in detecting bovine milk adulteration, even at low concentrations (as low as 2% (*v*/*v*)) [42]. Another study highlights the importance of N-acetyl carbohydrates as a significant biomarker for detecting bovine milk adulteration in caprine milk using ^1^H-NMR with multivariate data analysis (Figure 5) [55]. Moreover, the study conducted by Cui et al. [39] focused on the adulteration of ultra-high temperature processing (UHT) milk with reconstituted milk using NMR metabolite profiling. The study employed 1D and 2D-NMR experiments with chemometrics to develop a method for identifying UHT and reconstituted milk samples. Through their analysis, the authors identified L-carnitine, succinate, and acetate as biomarkers that could be used to detect the adulteration of UHT with reconstituted milk. Indeed, NMR metabolite profiling can be an effective tool for detecting possible adulteration. This is particularly relevant because UHT milk and reconstituted milk share similarities in their production processes, making it challenging to differentiate them using traditional methods [39].

A study conducted by Lachenmeier et al. [56] delved into the issue of adulteration of bovine milk-based infant formula with melamine, a compound that can have adverse health effects on infants. In order to detect melamine, the study compared the effectiveness of solution NMR and high-resolution magic angle spinning (HR-MAS). The experiments were performed on 400 MHz and 700 MHz NMR spectrometers, respectively. Results obtained from the NMR experiments were then compared with the most commonly used LC-MS/MS method. The study found that both NMR spectroscopy techniques were sensitive in detecting melamine adulteration in infant formula, thanks to the distinct singlet peak of the NH_2_ groups of melamine (resonating at 5.93 ppm), which do not overlap with other signals from milk metabolites. However, the reference LC-MS/MS procedure was the most sensitive, with LOD = 0.005 mg kg^−1^ (expressed as the amount of detected melamine per kg of infant formula), followed by HR-MAS and solution-state NMR (LOD = 0.69 and 33.3 mg kg^−1^, respectively) [58]. Nevertheless, the authors highlighted certain advantages of NMR and HRMAS over the LC-MS/MS procedure. These advantages include the minimal time required for sample preparation compared to the more complex preparation steps involved in LC-MS/MS and the possibility of performing non-targeted analysis. 

In another study, the ability of NMR to detect adulterated bovine milk powder samples was exploited by Bergana et al. [22]. The samples were collected worldwide and artificially spiked with the most common adulterants in milk, such as melamine, dicyandiamide, urea, sucrose, maltodextrin, ammonium sulfate, soy protein isolates, and whey protein concentrate. The study found that ^1^H-NMR combined with conformity index analysis (i.e., a statistical approach used to assess the level of adulteration in a sample) allowed for the detection of low concentrations of all adulterants (0.005–5% *w*/*w*), especially for melamine (0.005% *w*/*w*), which exhibited a distinct and characteristic signal at 5.95–5.96 ppm in the NMR spectrum [22]. The study concluded that NMR spectroscopy is a powerful and rapid method for ensuring authenticity and detecting adulteration in bovine milk powder.

Beyond high-resolution NMR applications in milk analysis, the use of low-resolution time domain NMR (TD-NMR) has gained attention for online milk processing and quality control in the dairy industry [12,97]. TD-NMR offers practicality, portability, and relatively low instrumental costs, making it suitable for industrial settings. Unlike high-resolution NMR, TD-NMR does not require cryogenic liquids and enables rapid routine NMR analysis in a non-invasive manner. This allows for online NMR analysis directly through the milk packaging, facilitating real-time monitoring and quality control during milk production. Soyler et al. [57] have effectively utilized a low-field benchtop ^1^H-NMR (43 MHz) instrument to observe the enzymatic hydrolysis of lactose in milk under continuous flow mode. NMR spectra of the reaction were acquired every seven minutes over a total period of 280 min, allowing the tracking of lactose consumption by monitoring the β-lactose peak. However, the signals from the α-lactose site, which include lactose, glucose, and galactose, were concealed due to signal overlap, and the signals from β-galactose, and β-glucose were hidden by the suppressed water signals. Despite those challenges posed by signal overlap at the low magnetic field strength, the researchers were still able to monitor the enzymatic reaction by exploiting the constant anomeric ratio between the α- and β-anomers of lactose throughout the monitored period [57]. This demonstrates the potential of benchtop NMR at the industrial level for online control of residual lactose concentration in lactose-free milk products, which is particularly relevant for meeting regulatory specifications regarding lactose content in milk.

In the field of milk adulteration and quality control, ^1^H TD-NMR has also been utilized. Santos et al. [58] applied ^1^H TD-NMR combined with T_2_ relaxation measurements and chemometric methods to detect adulteration in milk samples. The milk samples were artificially adulterated with common adulterants (water, whey, urea, etc.) at different concentrations in the range from 5% to 50% *v*/*v*. The adulteration level in milk samples could be detected through the T_2_ relaxation time, as higher concentrations of adulterated samples exhibited longer relaxation times compared to control samples. This trend was consistently observed across all milk samples adulterated with various adulterants at the same concentration level [58]. In another study, Coimbra et al. [59] successfully detected formaldehyde adulteration in bovine milk using ^1^H TD-NMR during refrigerated storage (0 and 48 h). The T_2_ relaxation curves showed that changes in relaxation times were directly proportional to the concentration of added formaldehyde, with higher concentrations of formaldehyde resulting in increased T_2_ relaxation times. The authors explained that the addition of formaldehyde caused coagulation of casein in adulterated milk samples during storage, leading to alterations in water and fat mobility and consequently increased T_2_ relaxation times. Such studies highlight the effectiveness and reliability of TD-NMR in the online assessment of milk quality and authenticity. 

## 9. Conclusions and Future Perspectives

This review focuses on the application of NMR spectroscopy in bovine milk analysis, specifically highlighting studies related to milk animal species origin, nutritional quality, geographical origin, and adulteration of bovine milk. NMR spectroscopy offers unique advantages in milk studies, including minimal to no sample preparation, rapid untargeted analysis, and the ability to identify unknown milk metabolites, which can serve as potential biomarkers for assessing the authenticity and quality of bovine milk.

The high limit of detection and the need for standardization of NMR methods are still considerable limitations compared to MS-based platforms, which are widely used in milk metabolomics. Nevertheless, ongoing advancements in NMR probe and magnet designs, magnet field strength, pulse sequences, and software and database improvements aim to address these limitations and position NMR as a complementary tool to MS platforms.

In milk research, it is recommended to adopt a multiplatform approach that leverages the strengths of each technique. A comprehensive study should employ NMR to achieve extensive metabolic coverage and high-throughput analysis. NMR can be combined with MS as a complementary method, considering its strength as a low-limit detection method. This sequential approach involves untargeted NMR analysis followed by the quantification of especially low-concentration milk metabolites that have been identified as important through NMR analysis.

Overall, NMR spectroscopy serves as a promising technique in milk studies, and with ongoing advancements, it holds great promise for further advancements in milk quality assessment and authentication research.

## Figures and Tables

**Figure 1 foods-12-03240-f001:**
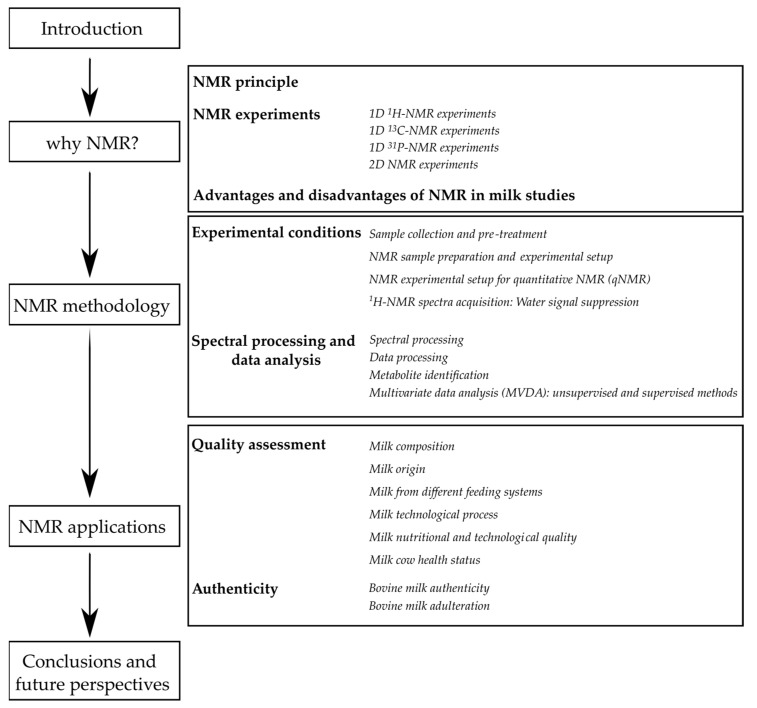
Workflow of the review.

**Figure 2 foods-12-03240-f002:**
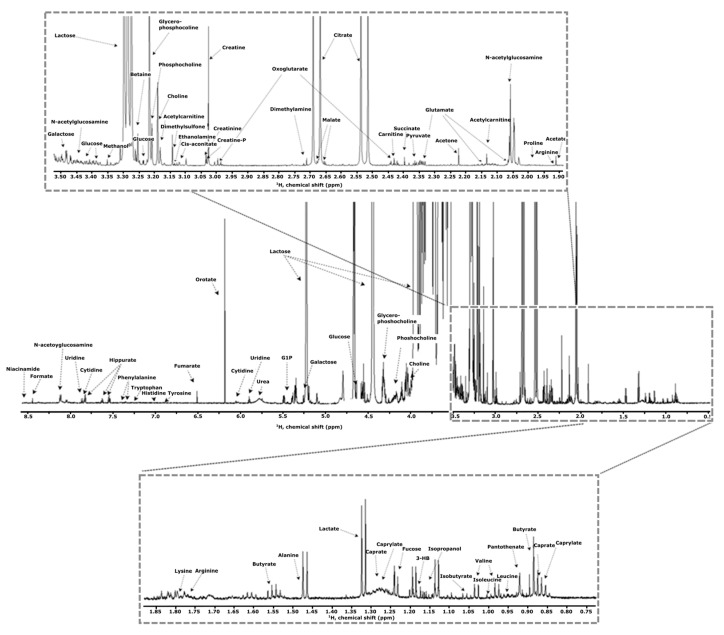
A representative 700 MHz ^1^H-NMR spectrum of commercial milk. The milk sample was dissolved in 10% D_2_O with 1 mM DSS as internal standard. The spectrum represents a comprehensive characterization of milk metabolites; it is dominated by the signals of carbohydrates (primarily lactose, glucose, and galactose; upper inset), organic acids (citrate), amine-containing compounds (creatine and urea; lower inset), orotic acid, and glycerophosphocholine. Additionally, amino acids such as tryptophan, tyrosine, histidine, isoleucine, leucine, and asparagine are also presented. Reprinted from [14] with permission from American Chemical Society**.**

**Figure 3 foods-12-03240-f003:**
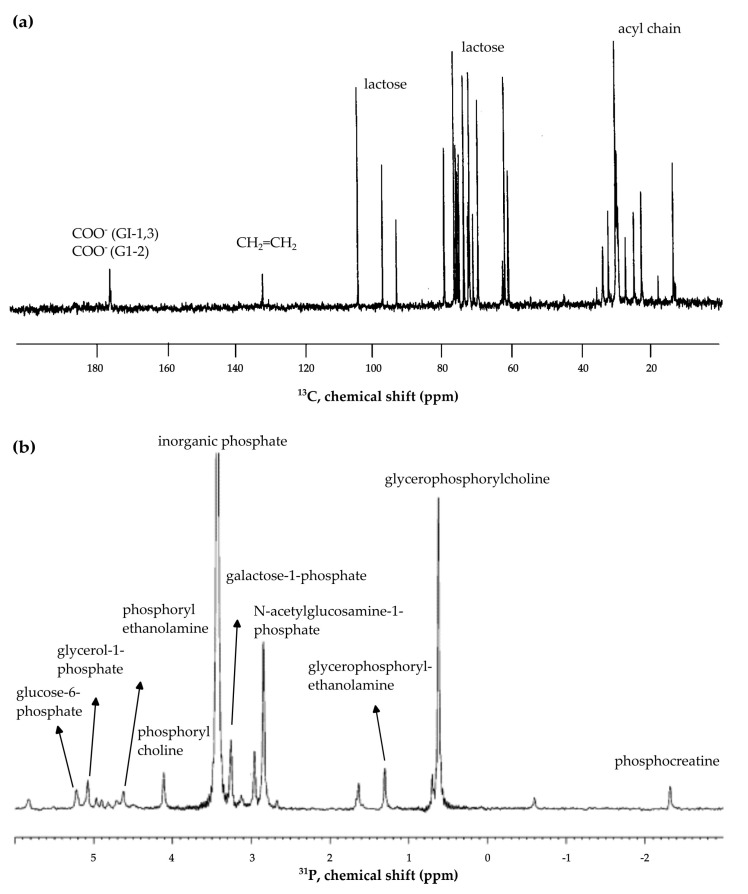
(**a**) 500 MHz ^13^C-NMR spectrum of whole milk in 10% D_2_O. The spectrum is dominated by the signals of lactose and acyl chains of lipids. Adapted from [36] with permission from American Chemical Society. (**b**) 400 MHz (operated at 161.97 MHz) ^31^P-NMR spectrum of milk ultrafiltrate obtained at pH 9.4 in triethylamine/dimethyl-formamide/guanidinium hydrochloride with 10% D_2_O and phosphocreatine as internal chemical shift reference. The spectrum represents the phosphorous-containing molecules in milk, with inorganic phosphate being the most abundantly present. Adapted from [37] with permission from Elsevier.

**Figure 4 foods-12-03240-f004:**
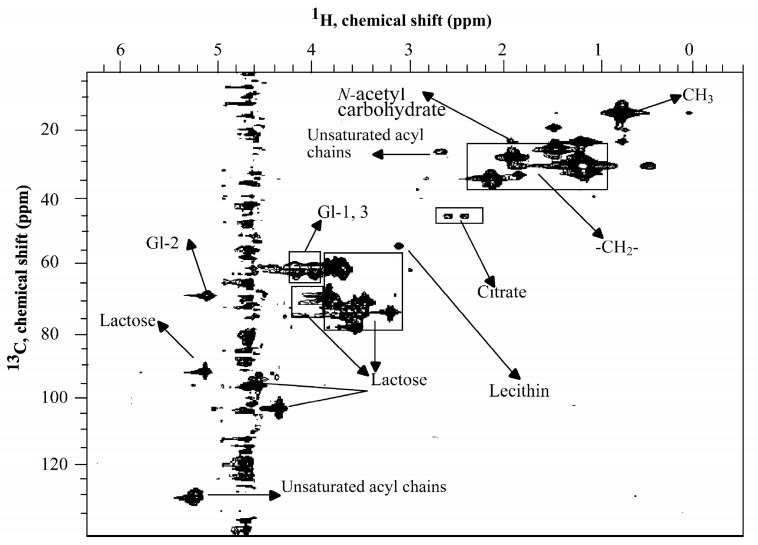
500 MHz 2D ^1^H-^13^C HSQC-NMR spectrum of whole milk in 10% D_2_O. The spectrum shows resonances from the main constituents of milk (mainly D-lactose, acyl chains of fatty acids, and glycerol backbone of fats) Adapted from [36] with permission from American Chemical Society.

**Figure 5 foods-12-03240-f005:**
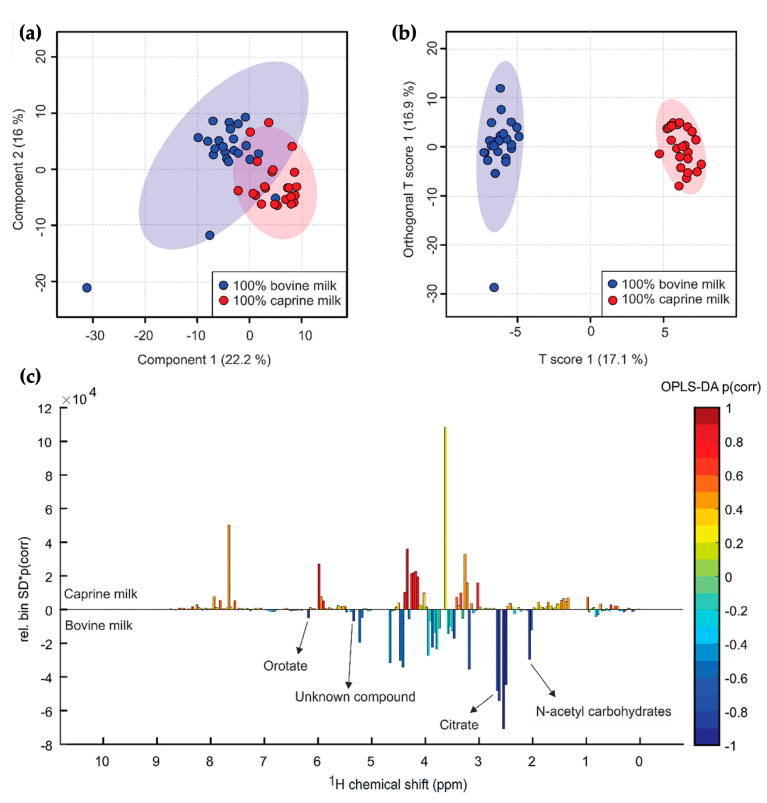
Example of a multivariate analysis procedure applied in NMR milk authentication studies. (**a**) PCA score plot; (**b**) OPLS-DA score plot; and (**c**) OPLS-DA corresponding color/coded coefficient plot of median ^1^H-NMR spectra of pure bovine (*n* = 23) and caprine milk (*n* = 23) represent the contribution of the metabolites toward class discrimination between bovine milk and caprine milk. The color of the bar indicates the correlation loading of the bin p(corr) value. The size of the bar was multiplied by standard deviation. Adapted from [55] with permission from American Chemical Society.

**Table 1 foods-12-03240-t001:** Advantages and disadvantages of NMR spectroscopy.

1D ^1^H-NMR		1D ^13^C-NMR		1D ^31^P-NMR		2D-NMR	
Advantages	Disadvantages	Advantages	Disadvantages	Advantages	Disadvantages	Advantages	Disadvantages
High sensitivity	Narrow spectral window	Relatively broader spectral window	Low sensitivity	High sensitivity and broad spectral window		Higher sensitivity than 1D ^13^C experiments	
Short relaxation time	Low resolution	High resolution	Long relaxation time	High resolution	Long relaxation time	high resolution, more detailed metabolite assignment,	
Short experimental time	Overcrowded spectra	Promising in milk fatty acid analysis	Long experimental time	Promising in milk phospholipid analysis	Long experimental time	Identification of unknown metabolites	Long experimental time
Suitable for metabolic fingerprinting		Suitable for metabolic profiling		Suitable for metabolic profiling		Suitable for structural elucidation	Usually not used for metabolic profiling

**Table 2 foods-12-03240-t002:** Summary of main characteristics of NMR studies on bovine milk ^1^.

Application Field	Sample Preparation	pH	NMR Experiment	Data Pre-Processing	Other Analysis	MVA	Reference
Chemical composition characterization	Whole and extract	na	^1^H, ^13^C-NMR, HSQC, HMBC		_	_	[36]
	Extract	7.0	^1^H-NMR	_	LC-RMS, LC-MS/MS ICP-MS	_	[14]
Feeding regime	Extract	na	^1^H-NMR	Integration	FT-IR	HCA, PLS-DA	[51]
	Extract	7.0	^1^H-NMR, TOCSY	Binning, integration	GC-MS	CDA	[17]
Milk quality and animal health	Extract	6.0.	^1^H-NMR	Binning	FT-IR	PCA, PLS-DA	[15]
	Extract	na	^1^H-NMR	Alignment, binning	Flow cytometry, MIR, microbiological analysis	PCA	[52]
	Extract	7.0	^1^H-NMR	Alignment, binning, integration	LC-MS	_	[53]
	Extract	7.0	^1^H-NMR	integration, normalization	_	rPCA	[16]
Technological process and shelf life	Extract of lyophilized milk	na	^1^H-NMR	_	UPLC-QToF/MS, GC-MS	PCA, PLS-DA, PLSR	[54]
	extract of lyophilized milk	7.4	^1^H, ^13^C-NMR	Integration	UPLC-QToF/MS	PCA, PLS-DA, PLSR	[19]
	Extract	7.0	^1^H-NMR	Normalization, alignment	_	PCA, PLS-DA	[20]
	Whole	na	^1^H, ^13^C-NMR	_	FT-IR, GC-MS, SDS-PAGE, microbiological analysis	_	[21]
Milk technological properties	Extract	na	^1^H-NMR	Alignment, binning	FT-IR, flow cytometry, rheometry	PCA, PLS, OPLS-DA	[45]
	Skimmed milk	na	^1^H, ^13^CNMR, ^1^H-^13^C HSQC	Alignment, binning	FT-IR, flow cytometry, rheometry	PCA	[46]
Infant formula quality assessment	Extract of infant formula	na	^1^H-NMR	Integration, normalization, binning	_	PCA, PLS-DA, OPLS-DA	[23]
Animal species origin	Extract	na	^13^C-NMR	Integration, normalization	_	Fuzzy logic analysis	[26]
	Extract	na	^31^P-NMR	_	_	_	[33]
	Extract	na	^31^P-NMR	Integration, normalization	_	PCA, UNEQ, SIMCA, K-NN	[31]
	Extract	na	^31^P-NMR	_	GC, LC-MS	_	[34]
Geographical origin	Extract	na	^13^C-NMR	_	IRMS	DA, SDA	[30]
	extract of lyophilized milk	na	^1^H-NMR, COSY, TOCSY	_	IRMS, HPIC, ICP-AES, FT-IR	PCA, DA	[43]
	Extract	7.4	^1^H-NMR, ^1^H-^1^H COSY, ^1^H-^13^C HSQC	Binning, integration, normalization	_	PCA, PLS-CA	[44]
Organic milk authentication	Extract	na	^1^H, ^13^C-NMR	Alignment, binning, integration	IRMS	PCA, LDA, FDA, PLSDA	[28]
	Extract of lyophilized milk	na	^1^H-NMR, 1D, 2D TOCSY, ^1^H-^13^C HSQC, HMBC	Binning, integration, normalization	_	PCA, PLS-DA	[29]
Bovine milk adulteration	Extract	na	^1^H-NMR, ^1^H-^13^C HSQC, HMBC	Binning	_	PCA, LDA, ANN	[18]
	Extract	na	^1^H, ^13^C-NMR, ^1^H-^13^C HSQC, HMBC	Integration, normalization	_	PCA	[42]
	Extract of reconstituted milk	7.4	^1^H-NMR, ^1^H-^13^C HSQC, HMBC	Integration, normalization	_	PCA, PLS-DA	[39]
	Extract	7.4	^1^H-NMR	Binning, integration, normalization	IR, flow cytometry, microbiological analysis	PCA, OPLS-DA	[55]
Infant formula fraud with melamine	Infant powder	7.0	^1^H-NMR, ^1^H HRMAS NMR	_	LC/MS/MS	_	[56]
Milk powder adulteration	Extract of milk powder	na	^1^H-NMR	Binning, normalization	_	_	[22]
Online quality and adulteration control	Whole and skimmed	na	Benchtop ^1^H-NMR	Integration, normalization	_	ANN	[57]
	Whole milk	na	^1^H TD-NMR	_	_	PCA, SIMCA, k-NN, PLSR	[58]
	Whole milk	na	^1^H TD-NMR	_	Colorimeter	PCA, PLS, SIMCA	[59]

^1^ Extract represents the sample extraction process through sample preparation. Abbreviations are: HSQC, heteronuclear single quantum coherence spectroscopy; HMBC, heteronuclear multiple bond correlation spectroscopy; TOCSY, total correlation spectroscopy; COSY, correlation spectroscopy; HRMAS, high-resolution magic angle spinning; MVA, multivariate statistical data analysis; PCA, principal component analysis; DA, discriminant analysis; UNEQ, unequal class modeling; SIMCA, soft independent modelling of class analogy; K-NN, kernel nearest neighbor; HCA, hierarchical cluster analysis; PLS-DA, partial least squares DA; CDA, canonical DA; PLSR, PLS regression; OPLS-DA, orthogonal PLS-DA; ANN, artificial neural network; rPCA, robust PCA; LDA, linear DA; FDA, factorial DA; SDA, stepwise DA; LC-HRMS, liquid chromatography high-resolution mass spectrometry; LC-MS, liquid chromatography mass spectrometry; ICP-MS, inductively coupled plasma mass spectrometry; GC, gas chromatography; FT-IR, Fourier transform infrared spectroscopy; UPLC-QToF/MS, ultraperformance liquid chromatography-quadrupole time-of-flight MS; SDS-PAGE, sodium dodecyl sulphate-polyacrylamide gel electrophoresis; MIR, mid-infrared spectroscopy; IRMS, isotope ratio MS; HPIC, high-performance ion chromatography; ICP-AES, inductively coupled plasma atomic emission spectroscopy; na, not adjusted.

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
