# Peer review of "Applications of Solution NMR Spectroscopy in Quality Assessment and Authentication of Bovine Milk"

_foods, 2023, doi:10.3390/foods12173240_

Round 1

Reviewer 1 Report

The present manuscript demonstrates a systematic and very comprehensive review on the applications of Nuclear Magnetic Resonance (NMR) spectroscopy in milk metabolomics research. Information on the principles of NMR detection, sample preparation, spectral and data processing have been provided. Several papers focusing on the NMR application in the investigation of milk composition, technological quality, authenticity and traceability have been highlighted with their advantages. The manuscript was very well prepared and written with a clear English language level. In my humble opinion, however, several minor suggestions as indicated below should be clarified.

·       Line 10-22: The advantage of NMR as “a tool for milk and dairy metabolomics research” is supposed to being mentioned here.

·       Line 22: I would suggest replacing the word “ensuring” by “characterizing or assessing” instead.

·       Line 247-278: The limitation of 2D-NMR should be also mentioned. Why this technique is less employed in milk metabolomics, along with a lower number of publications, compared to 1D-NMR?

·       Line 299-230: I would suggest using “minimal sample preparation steps” (see line 312) since in most milk metabolomics studies, the separations of milk lipids and proteins are still required before subjecting samples, e.g. milk serum or extract, to the NMR analysis.

·       Line 37: I do totally agree with this statement.

·       Line 327: Table 1 is a good summary.

·       Line 544-590: An additional paragraph referring to steps in metabolite identification as well as available databases should be described.

·       Line 624: Incomplete sentence. Ref [66] should be added.

·       Line 647; Section 7: Information in Table 2 should be somehow mentioned when appropriate here.

Reviewer 2 Report

The manuscript can be interesting but needs some revision before further processing.

The background of this work is not much clear. What is the novelty of this work that should be clearly stated?

Add schematics to show the overall work

Information related to NMR is well known, so better to make it concise

Figure quality is very poor, improve it carefully

Add a Table in the Application section.

Also authors are encouraged to add some schematic diagrams in the application part.

Use up-to-date citations and remove old ones.

Also carefully correct the linguistic and typos errors.

Reviewer 3 Report

The authors described the applications of NMR spectroscopy in the quality assessment and authentication of bovine milk. My comments are as follows:

1. There are so many typos in the main text.

2. As a scientist using MASS, the reviewer acknowledges the advantages of NMR spectroscopy you claimed. However, please show me a Table that points out the basis or concept and compares them.

3. Why did the authors confine themselves to bovine milk? I think readers will also be very curious about breast milk and/or camel milk.

 Extensive editing of English language required.

Round 2

Reviewer 3 Report

I enjoyed the revision you sent me; it was well written. I have one more thing. Do you think it is necessary to compare the NMR method currently expressed in the Introduction section with various competitive methods (MALLS, FPLC-GPC, HIC, ESI-MS, MALDI-MS, etc.) and express the advantages and disadvantages of each of them?

Extensive editing of English language required. 

Author Response

(Please see the attachment)
